# A Systematic Review on Machine Learning and Deep Learning Models for Electronic Information Security in Mobile Networks

**DOI:** 10.3390/s22052017

**Published:** 2022-03-04

**Authors:** Chaitanya Gupta, Ishita Johri, Kathiravan Srinivasan, Yuh-Chung Hu, Saeed Mian Qaisar, Kuo-Yi Huang

**Affiliations:** 1School of Computer Science and Engineering, Vellore Institute of Technology, Vellore 632014, India; chaitanya.gupta2019@vitstudent.ac.in (C.G.); kathiravan.srinivasan@vit.ac.in (K.S.); 2School of Information Technology and Engineering, Vellore Institute of Technology, Vellore 632014, India; ishita.johri2019@vitstudent.ac.in; 3Department of Mechanical and Electromechanical Engineering, National ILan University, Yilan 26047, Taiwan; ychu@niu.edu.tw; 4Electrical and Computer Engineering Department, Effat University, Jeddah 22332, Saudi Arabia; sqaisar@effatuniversity.edu.sa; 5Department of Bio-Industrial Mechatronic Engineering, National Chung Hsing University, Taichung 402, Taiwan

**Keywords:** network, information security, cyber security, artificial intelligence, machine learning, deep learning, threats, cyber-attacks, vulnerabilities

## Abstract

Today’s advancements in wireless communication technologies have resulted in a tremendous volume of data being generated. Most of our information is part of a widespread network that connects various devices across the globe. The capabilities of electronic devices are also increasing day by day, which leads to more generation and sharing of information. Similarly, as mobile network topologies become more diverse and complicated, the incidence of security breaches has increased. It has hampered the uptake of smart mobile apps and services, which has been accentuated by the large variety of platforms that provide data, storage, computation, and application services to end-users. It becomes necessary in such scenarios to protect data and check its use and misuse. According to the research, an artificial intelligence-based security model should assure the secrecy, integrity, and authenticity of the system, its equipment, and the protocols that control the network, independent of its generation, in order to deal with such a complicated network. The open difficulties that mobile networks still face, such as unauthorised network scanning, fraud links, and so on, have been thoroughly examined. Numerous ML and DL techniques that can be utilised to create a secure environment, as well as various cyber security threats, are discussed. We address the necessity to develop new approaches to provide high security of electronic data in mobile networks because the possibilities for increasing mobile network security are inexhaustible.

## 1. Introduction

Electronic information is an asset for any organisation, and even in the case of an individual, their data can be quite significant to them, which they cannot afford to lose. Information security has become very important in today’s computing world, and it demands potential counters to ever-evolving threats. Hence, cyber security and risk management are vital for data-driven or information-dependent tasks. 

Cyber security is a collection of procedures, actions of people, and technology that aid in the protection of electronic information resources. It is obvious that cyber attackers are outpacing defences, raising concerns about the security of sensitive digital assets [1]. The statistics on vulnerabilities and unauthorised access show most information-sharing devices, especially mobile networks, are at a high-security risk. The first stage in evaluating a system’s security or assessing risks is to identify resources. It is very important to identify a comprehensive approach that is best suited to the risky situation. This aids in the adoption of the most cutting-edge method for predicting information security threats and their subsequent mitigation. The best-suited model may also depend on the attack scenarios and target of attack. Hence, proper research is required to deal with electronic information security. As the number of cyber-attacks, particularly on mobile networks, increases, so does the performance of our systems to combat them, as discussed in the following sections. 

Figure 1 depicts the general taxonomy of artificial intelligence techniques. There exists an extensive horizon of trans-disciplinary association between cybersecurity and artificial intelligence. Technologies such as deep learning could be deployed in devising sophisticated models for intrusion detection, malware classification, and cyber threat sensing in a mobile network. AI models require specialised cybersecurity and protection solutions to minimise vulnerabilities and ensure better privacy of information, as well as to enable a safe federated learning environment [2]. The development of artificial intelligence has led to the emergence of many other fields such as ML, NLP, computer vision, etc. [3,4,5,6,7,8,9,10,11,12,13,14,15,16,17,18,19,20,21,22,23,24,25]. Deep learning, one of the most prominent subsets of AI, is making significant progress in solving challenges related to information security threats [26,27,28,29,30,31,32,33,34,35,36,37,38,39,40,41,42,43,44,45,46,47,48,49,50,51]. The models built using these technologies work within IT risk management frameworks to create a secure ecosystem of networks.

Mobile ransomware, crypto mining, fraudulent apps, and banking Trojans are among the most common dangers to mobile networks. Mobile applications have surpassed desktop programmes as the most popular method of accessing personalised services such as sending and receiving emails, banking services, online shopping, and automated device controls. Hackers have taken advantage of the patch systems to infect mobile apps, making them ideal targets for cybercriminals. Existing security solutions appear to be insufficient for the impending mobile technology, which has increased transmission rates on networks. With improvements in AI technology, complex models are making breakthroughs in the security of various critical applications, many of which are based on mobile networks. However, this does not mean that the capabilities of threats exploiting our system have been reduced. The extensive advancements in the mobile network domain help the new generation of networks to be much faster and more secure than the previous versions. Still, the challenge to security from identified and unidentified risks indicates the need for an extension of ongoing risk management systems. As a result, while having a profusion of frameworks for securing an organisation’s resources from cyberthreats, the option for cyber security decision makers remains mostly challenging [3]. 

We propose to offer a comprehensive analysis of the main approaches that can be employed for the security of electronic data based on observations made during the systematic revision and assessment of AI-enabled technique solutions. This article gives a complete review of how and why AI technology has been employed for electronic information security, as well as list of the mobile network security domains where AI-enabled techniques have been used. A thorough examination of existing ML and DL models was conducted in order to gain a clear understanding of how this technology contributes to vulnerability detection and mitigation. We can also figure out where these models would be vulnerable to a cyber-attack. Because most of us now spend a significant amount of time on our mobile devices, the data we generate or use is subject to a variety of cyber-attacks, which will be discussed in more detail in subsequent sections. It has also been addressed in relation to a number of popular datasets that help in the creation of secure architecture for networks. The subject is better understood with the help of many figures and tables. Finally, this paper encourages discussion of potential research topics and open issues related to information security in mobile networks. Appendix A contains a list of acronyms used in this review, as well as their definitions.

### 1.1. Contribution of This Survey

This paper bridges the gap between artificial intelligence and electronic information security by offering a comprehensive assessment of machine learning and deep learning approaches and techniques for electronic information security in mobile networks. Table 1 presents the comparison of this work with the previous reviews. The key contributions of this study are summarised as follows:We provide a comprehensive study on the various machine learning and deep learning models used for electronic information security in mobile networks. A brief explanation of several machine learning and deep learning methodologies is included.A concise review of cyberattacks as well as an application-oriented analysis of their datasets is given.We highlight the current open challenges and future research possibilities in the fields of mobile networks, electronic data security, and cyber threats for aspiring researchers and enthusiasts to investigate.

### 1.2. Survey Methodology

We used the “preferred reporting items for systematic reviews and meta-analyses (PRISMA)” methodology to select the papers for this study.

#### 1.2.1. Search Strategy and Literature Sources

From January 1998 to January 2022, articles on developing metaheuristic algorithms were searched on Google Scholar, ScienceDirect, IEEE Xplore, ACM Digital Library, IET Digital Library, Wiley Online Library, Springer Nature, and Springer databases. Using search terms such as “Electronic Information Security” or “Cyber Security models” or “Mobile Network Security” or “Cyber-attack” or “Artificial Intelligence” or “Machine Learning” or “Deep Learning Security model”, this analysis found roughly 1800 plus articles.

#### 1.2.2. Inclusion Criteria 

This study includes articles about electronic information security in mobile networks that were authored and published in the English language between January 1998 and January 2022. This review contains some recent findings that help in taking research work one step forward.

#### 1.2.3. Elimination Criteria

This review excluded reports, case studies, editorials, publications, analyses, conference proceedings, doctoral dissertations, and theses that were published in languages other than English or before January 1998.

#### 1.2.4. Results

Initially, duplicates were removed from 1880 publications, and 780 were chosen for a full text analysis after assessing their abstracts. The investigation includes journal and conference articles. After reading the full text of these publications, 603 are eliminated because they were using duplicate methods or had previously been published. In the end, 177 publications were examined in this study. Using a PRISMA diagram, Figure 2 depicts the process of selecting papers for this study.

### 1.3. Structure of this Survey

Figure 3 depicts the layout of this survey paper. Section 1 defines the terms “Electronic information Security” and “Mobile networks” and explains how this field has grown in recent years. It also discusses existing technologies that are already being used for the same purpose, as well as how we performed our study to proceed. Section 2 provides an in-depth look at a variety of AI-enabled methodologies, with a focus on machine learning and deep learning models. Several cyber-attacks and their defence mechanisms are detailed in Section 3. It also provides an overview of several well-known datasets that can aid in the development of a robust cyber-attack defence mechanism or model for mobile networks. Section 4 discusses open information security challenges. In Section 5, opportunities and various future research directions are mentioned. Finally, Section 6 brings the paper to a close. At the end of the paper, several reference materials used in its preparation are listed.

## 2. Machine Learning and Deep Learning Models Used in Electronic Information Security Applications

### 2.1. The Evolution and Overview of Machine Learning and Deep Learning Models

AI is an area of computer science that deals with intelligent systems. It is a fast-growing field, and it has been developing rapidly in the last few years. The evolution of AI is an ongoing process, and we can expect to see more developments in the future. AI is being developed to help humans in many ways, from decision-supporting to helping with the most mundane of tasks. The early days of AI saw the use of expert systems, which were programmed to provide advice and decision support to human users [1]. These expert systems were developed using rule-based programming languages such as Prolog, LISP, etc. The next stage was natural language processing (NLP), which helped machines understand human speech better by converting words into computer-readable text [52]. This made it possible for computers to understand written or spoken language as well as humans do.

In this section, we will explore the history of AI and the future of computing power. We will also see what developments are taking place in the field of security models for mobile networks, what human decisions are being supported by AI, and what these systems can do for us today.

### 2.2. Machine Learning Techniques

Machine learning is a data-driven approach to developing artificial intelligence. It is a subset of AI, and it has many strengths and uses statistical techniques for predictive purposes [53]. It was developed in the 1940s, but it wasn’t until recently that we have been able to use it in our everyday lives. ML algorithms generally use the following two types of learning methods: supervised and unsupervised learning. Unsupervised learning requires no feedback, whereas supervised learning relies on human feedback [52]. ML has many strengths, but the most important ones are its methodology and reinforcement learning. The ML methodology involves training an algorithm with a set of data so it can identify patterns in new data sets [54]. Reinforcement learning is a class of ML where the intelligent system learns by trial and error by being rewarded or punished for its actions. In the following section, we will be reviewing the machine learning models for electronic information security.

#### 2.2.1. Artificial Neural Network

Artificial neural networks (ANNs) are based on the biological neural networks seen in the human brain and are analogous to them. They are built up of artificial neurons that are interconnected and interact with one another [42]. ANNs are made up of two visible layers (an input layer and an output layer) as well as one or more hidden layers. The input layer is where information comes into the network from the outside world, and the output layer is where it leaves as a result of processing. In between these two layers, there are hidden layers that consist of nodes that receive input from nodes in the previous layers and send signals to nodes in the next layers. These hidden layers help the neural network process information in parallel computations [1].

The first artificial neural network was created in 1943 by Warren McCulloch and Walter Pitts. They were trying to model the human brain and how it learns through feedback. ANNs have the ability to learn by adjusting weights for each neuron based on feedback. The hidden layers have many parameters, called weights, which can be adjusted during training to increase or decrease how much influence a particular neuron has over its neighbours. They also use backpropagation, which adjusts weights based on error values, to make sure that they minimise errors.

Dimitrios Damopoulos et al. [55] used exploratory methods to forecast and identify probable unauthorised activities and undesirable occurrences in user behaviour in terms of phone calls, text messages, web browsing history, and multimodal data. Bayesian networks, RBF, KNN, and Random Forest were all tested as supervised machine learning algorithms. RBF is a feed-forward network with a single hidden layer that is commonly used to solve approximation problems. The final goal was to construct frequent-usage mobile user behaviour profiles with the goal of informing users when their behaviour deviates from the norm.

Before they achieved their intended target, Saied et al. [56] employed ANN to detect and mitigate predictable and unpredictable DDoS attacks (TCP, UDP, and ICMP protocols). The DDoS attack protocols, TCP, UDP, and ICMP, were chosen based on their prevalence among DDoS attackers.

#### 2.2.2. Naïve Bayes

Naive Bayes fall under the category of probabilistic classifiers, which are used to classify data based on the data’s probability distribution and characteristics. The NB model implies that the features are independent, which means that the likelihood of a feature being present is unaffected by the presence or absence of other features. This assumption simplifies the computation by ignoring conditional probabilities. Naive Bayes assumes that the conditional distributions for each class are independent given the model parameters, or equivalently, that the joint distribution of all classes can be factored as a product of independent factors [53]. This means that Naive Bayes ignores any dependencies between the features when estimating the probabilities of a label in a classification problem and any correlations between features when estimating regression coefficients. The naive assumption simplifies the computations because it avoids having to explicitly store or update covariance matrices in memory.

In 2014, F. A. Narudin, A. Feizollah, presented a study on identifying mobile malware using several machine learning algorithms [6]. Two datasets are used in the following evaluation: public (MalGenome) and private (self-collected). The Bayes network had a 99.97 percent true-positive rate (TPR) on the MalGenome dataset, compared to a TPR of only 93.03 percent for the multi-layer perceptron.

In terms of telephone conversations, SMS, and web browsing history, the Bayesian networks provided highly promising results, with an average TPR and accuracy of 99.06 percent and 98.76 percent in the worst circumstance, respectively, as proposed by Damopoulos et al. [55].

#### 2.2.3. Decision Tree

Decision trees are a popular data analysis technique for classification and regression problems. A decision tree is a recursive tree structure that helps to classify or predict values of a target variable. The recursive tree structure is built by repeatedly splitting the data set into the following two parts: one with the members that satisfy the current test, and one with those who do not satisfy it [1]. The root node is the starting point for the tree, and it has branches that lead to other nodes, which are called terminal nodes. These terminal nodes are the leaves of the tree, which represent different observations or outcomes of the target variables [57].

Decision trees work by checking the data against an entropy measure to determine the best split for each node. The entropy is calculated by counting the number of bits needed to encode the input and then dividing it by two. The best split is found by calculating information gain, which is calculated as the following: gain = how much entropy we removed.

Decision trees work by calculating entropy and information gain for each split point in order to decide which variable should be used to divide the data. The entropy calculation is used to measure how evenly distributed our data is; meanwhile, information gain calculates how much information will be gained by using one. The advantage of decision trees is that they provide an easy way to visualise how data is classified into groups by splitting it at different points in order to find patterns in the data. 

Wu et al. [58] proposed a DDoS-detection model based on decision-tree techniques (C4.5). It also included a new attacker traceback based on GRA traffic pattern matching (grey relational analysis). With separate attacks, the false positive ratio was about 1.2–2.4 percent and the false negative ratio was about 2–10 percent while performing the experiment to detect DDoS attacks.

#### 2.2.4. K-Nearest Neighbour

The K-nearest neighbour is a supervised learning algorithm that is used to assign objects into one of the K-clusters. The algorithm assigns each object to the cluster with the closest mean, and it is one of the simplest algorithms for clustering. It works by finding the K-nearest neighbours to a given point and then classifying the point as belonging to whichever of those k points is most similar. The KNN algorithm classifies data points into different clusters by comparing their features to the features of the other data points in their cluster. The number of clusters can be determined by setting a threshold on the similarity values between two points [12]. This threshold is often set at 0.5, meaning that two points are put into different clusters if they are more than half as similar as another point in their cluster. The number of neighbours to use for classification can be determined by setting a parameter called k, which represents how many neighbours to compare with each point before making a decision about its cluster membership [1].

The K-nearest neighbour algorithm has the following number of advantages:It does not require any assumptions about the shape or distribution of input data.It does not require any pre-processing or transformation of input data before assigning them to clusters, making it computationally effective. It does not require any parameter tuning and can be used for both dense and sparse sets of data.

KNN is a lazy learner in the sense that it doesn’t update its model after it makes predictions as compared to K-means, which is an eager learner. It just remembers the last prediction made and then predicts the same thing again when asked to make a new prediction. 

In the study published by Narudin et al. [6] on detecting mobile malware effectively using different ML techniques, the KNN classifier achieved the highest accuracy of 84.57%. Moreover, the greatest precision, recall, and f-measure value were produced by KNN.

#### 2.2.5. K-Means Clustering

Clustering is a powerful technique for exploring data because it groups together items that are similar to one another. K-means clustering is an iterative machine learning algorithm that is used for finding clusters in data [42]. It is simple to implement, has a low computational cost, and can be applied to both continuous and discrete data. It starts with the random selection of k values. These values are called centroids, and they are used to create clusters. K-means clustering has the disadvantage of requiring the specification of an arbitrary number of clusters, which can be difficult to determine in advance. The higher the k value, the more evenly we split our data points across clusters. The lower the k value, the more tightly packed our data points will be in each cluster. The algorithm then assigns each data point to the closest cluster by calculating the distance between each point and every centroid. The next round of clustering is performed by assigning new centroids to the clusters that were created in the previous round.

In the experiments carried out by Do et al. [59], three alternative initiation methods were used to test the K-means classifier. K-means is a cluster classifier that works best when clusters are evenly distributed, which was not the case with DNS tunnelling. Where only a small portion of the traffic data was malicious, K-means still provided quite interesting results, but was unable to detect outliers for the given dataset.

#### 2.2.6. Random Forest

Random forest is a type of ensemble learning technique for classification and regression. RF is a supervised machine learning algorithm used to build classification or regression models by combining many decision trees, each of which is built using only part of the data [6]. The technique was first proposed by Leo Breiman in 2001 as a way to reduce variance in predictions from decision trees because it gives equal weight to each tree’s prediction and does not over-rely on any one particular tree that may have been overfit to the training data used to create it. Each decision tree in the forest is independent of the others, so different trees may give different classifications for a particular input. The final classification or regression model is based on a majority vote of each individual decision tree [53].

The Random Forest algorithm is often used for classification and regression problems, but can also be applied to clustering and other tasks. It has been shown to outperform both individual decision trees and other ensemble methods in many cases, while being significantly easier to fit.

Belavagi et al. [60] attempted to compare the performance of supervised machine learning classifiers such as SVM, Random Forest, logistic regression, and Naive Bayes for building intrusion detection systems. For the dataset (NSL-KDD) and parameters evaluated, the Random Forest classifier outperformed other classifiers as it possessed an accuracy rate of 99 percent based upon the precision, recall, F1-Score, etc.

Random Forest produced the best results in Damopoulos et al. [55]’s performance evaluation on telephone conversations, SMSs, and web browsing histories, with its average TPR and accuracy standing above 99.8% and 98.9%, respectively, in all circumstances.

#### 2.2.7. Support Vector Machine

An SVM is a supervised learning model that is used to classify data into different classes. SVM is a classification algorithm that was created by Vladimir Vapnik and Alexey Chervonenkis in 1963. It can prove to be useful when creating a classifier for any given data set. An SVM works by mapping the data points onto a higher-dimensional space, called the feature space, and then separating them using a hyperplane. Support vector points are data points that are located on the hyperplane’s edge.

SVMs are classified into two classes based on the kernel function, which can be either linear or nonlinear, and the detection type, which can be either one-class or multi-class [12]. The hyperplane is chosen so that it divides the classes as equally as possible while maximising the distance between them. SVMs model the decision boundary in the space of possible inputs by a hyperplane that optimises some measure of “margin”. The hyperplane is then parameterized with a kernel function that defines the distance from the hyperplane to points in its vicinity. 

In mobile networks, Do et al. [59] proposed using OCSVM with an RBF kernel to identify DNS tunnelling. Four clients were used to originate and gather both benign and malicious DNS traffic in a testbed. Experiments were conducted, with the results demonstrating OCSVM poly kernel, a viable technique for detecting DNS tunnelling with the highest F-score measurement.

#### 2.2.8. Ensemble Models 

A recent trend in machine learning is ensemble models. These models are created by combining multiple classifiers to form a more accurate prediction. Ensemble models have the advantage of being able to be trained on a variety of datasets. This allows them to be more accurate than individual classifiers since they have more information to work with. Ensemble models also have the ability to handle new data better than single classifiers because they are not as specialised as single classifiers are. Ensemble models are created by training many different classifiers on the same dataset, and then combining their predictions. This is performed because each individual classifier has its own strengths and weaknesses [42]. For example, some may be better at differentiating between classes than others, or they may be better at predicting something other than the final classification (such as the probability of an intrusion). By combining their predictions, ensemble models can take advantage of each individual classifier’s strengths while also mitigating their weaknesses.

The ensemble model has been found to be more accurate than any individual model in some cases. There are the following two types of ensemble models: bagging and boosting. Boosting is the most popular type of ensemble model because it overcomes many weaknesses that were found in bagging models. The most common type of ensemble model is called an “ensemble method,” which is composed of decision trees or boosted decision trees. 

Based on the concept of stacking, Rajagopal et al. [61] suggested an ensemble approach for effective network intrusion detection. The stacking method was used to improve the accuracy and F measure of malware detection on mobile devices. A mixture of techniques, including random forest, logistic regression, K-closest neighbour, and support vector machine, gave better predictions on a real-time dataset than on an emulated dataset. Among the 7 known attacks detected in the UGR’16 dataset, the suggested ensemble model was the most effective in detecting the occurrence of the blacklist attack type. Such an attack detection potential, when demonstrated by intrusion detection models, can be effective in addressing new threats such as DDOS, DOS, and scan attacks.

#### 2.2.9. Machine Learning Models for Electronic Information Security

Network security is an ever-changing field with new threats popping up every day. Machine learning models provide the opportunity to stay up-to-date with the latest technological advancements in this field, while providing effective protection against these new threats as well as the old ones that have been around for years. In this section, we discussed some of the other machine learning models that can also be used in cybersecurity, such as intrusion detection and anomaly detection. Intrusion detection is a process where an intrusion is detected and prevented before it can cause any damage to the system. Anomaly detection, on the other hand, focuses on identifying anomalies in a system. These techniques help identify malicious attacks by looking for unusual patterns or behaviours. Biologically inspired methods are commonly used in other machine learning algorithms, but research on the artificial defence system is limited.

Software defined network-based (SDN-based) detection systems built on machine learning and/or implemented in the cloud have been proposed to guard networks from DDoS attacks, which convert targeted virtual computers (as a replication of users’ PCs) to safe virtual machines. [39]. Another growing subject in machine learning is semi-supervised learning, which is defined as a combination of supervised and unsupervised learning [62]. Reinforcement learning models for network security have been developed to help with the detection and prevention of cyber-attacks. These models are based on the idea that a system should be able to learn from its past experiences and then use this knowledge to make decisions in the future. It can be trained to detect certain patterns or behaviours that are indicators of an attack. The learning model will then send out alerts when it finds something suspicious, so that humans can take over if necessary and investigate further.

Figure 4 illustrates the nomenclature of current machine learning models for electronic information security. Table 2 presents the summary of works on machine learning techniques for electronic information security.

### 2.3. Deep Learning Models

Typically, in terms of performance, the deep learning models have surpassed the generic machine learning approaches. These DL models have a deep structure and possess automatic learning capabilities. Moreover, they generate an output without any external intervention. Unlike traditional machine learning approaches, deep learning models use layers of multiple artificial neurons to work efficiently at a level that can learn extremely complex tasks without being explicitly programmed [73]. 

In the past, neural networks have been used for tasks such as facial recognition or voice recognition by analysing images or sound waves, respectively. In the past 2 years, however, these neural networks have started to take over many other roles in our society, and society will never be quite the same again. In this section we present the deep learning models for electronic information security especially in mobile networks. Table 3 presents a summary of works on deep learning models for electronic information security.

#### 2.3.1. Recurrent Neural Networks

Recurrent neural networks (RNNs) are a type of neural network that is made up of many layers. They are designed to be able to process sequential information. The recurrent connection in the network allows it to store information in its memory cell and use it when processing new data [74]. They also have parameter sharing, which makes it easier for the network to learn and memorise things but harder to train at the same time [53]. These models also use gradient descent as a method for learning and memory cells as a way of storing information. 

Information security in mobile networks is one of the applications of RNN. It is capable of intrusion detection, network analysis, and even modelling the time series profile of LTE networks. Ailing Xiao et al. [75] developed a quality of experience (QoE) prediction engine based on a recurrent neural network (RNN) that was able to account for the following three descriptive parameters: video clarity, rebuffering, and rate adjustment in two phases, as well as the impact of several cognitive factors in a mobile network. The model accounts for quality mobile network experience and data management. 

#### 2.3.2. Deep Autoencoder

The deep autoencoder algorithm is an unsupervised learning algorithm that learns to encode input data into a lower-dimensional space [74]. Encoding and decoding are the two stages of the deep autoencoder’s architecture. The encoding stage is where the input data is transformed into a lower-dimensional space by applying a series of linear transformations to the input data and then passing it through an activation function [27]. The decoding stage takes the compressed representation and reconstructs it back to its original form, with each layer in the network performing a different transformation on the input data (e.g., adding more features). They have a number of parameters called “hidden units” that are learned from previous training data. 

This type of neural network relies on artificial intelligence and has been found to be both computationally and memory-efficient in its interpretation of data from sensory input such as images, video, or sound waves. They come with a large variety of applications, including natural language processing, image recognition, identification, and compression. Nasir Rahim et al. [76] have come up with a mechanism that ensures a user’s privacy when information in the form of images is being communicated through mobile networks. Images are represented as hash codes, which are compressed representations of deep convolutional features generated by an auto-encoder in the Cloud. 

Since the mobile phone does not support ultra-wideband and the positioning cost is expensive, ultra-wideband positioning is difficult to achieve a wide range of inside coverage. Due to the sheer uncertainty of the mobile phone’s initial orientation and the accuracy of the inertial sensor, inertial navigation is not ideal. Yanru Zhong et al. [77] developed a deep neural network using stacked auto encoders and performed pre-training to obtain a more accurate Wi-Fi indoor positioning model.

#### 2.3.3. Long Short-Term Memory

The LSTM architecture is an element of recurrent neural networks that was created to solve the problems that conventional feedforward networks faced. As they let computers interpret phrases with numerous layers of meaning, LSTMs are frequently used in natural language processing. Long short-term memory (LSTM) is capable of remembering values for much longer than the typical artificial neuron. The LSTM architecture was designed to address the vanishing gradient problem in traditional neural networks, which occurs when each connection within a network carries an independent weight that does not change over time [74].

The LSTM can learn “context” by storing information about what happened before and after every sequence it observes. This makes it possible for the LSTM to make sense of data that has been split up into separate, unrelated pieces. Forget gates are one of its main components. These gates control how much information is stored in the network, and they are usually set to one for simple tasks. In order to train an LSTM, it needs a lot more data than a typical neural network as well as more computational power. In addition, the training process can take significantly longer than for other types of deep learning networks. 

Many applications of LSTM have been successfully implemented, including wearable activity recognition, text categorization, IDS classifier, and so on, which deal directly with information security. It can also be used to detect anomalies in a 5G network [78]. Qingshan Wang et al. [79] proposed a supervised deep learning system to convert opportunistic forwarding to fixed path forwarding in data package communication. LSTM (Long Short-Term Memory) effectively improves packet delivery ratio while drastically lowering network overhead in this case. It can be used to investigate the data’s hidden characteristics and rework data forwarding techniques.

#### 2.3.4. Deep Neural Network

Deep neural networks, unlike shallow neural networks, feature multiple hidden layers between the input and output layers, resulting in a huge number of hidden layers [73]. The number of these hidden layers determines the network’s depth. The input layer takes in data from outside sources and processes it through the rest of the network. The middle layers are called “hidden” because they cannot be seen by looking at the network itself. The last layer is called “output” because it’s where all of the processed data comes out after being processed by all of these layers. Deep artificial neural networks, in contrast to shallow artificial neural networks, have been demonstrated to be capable of learning features at multiple levels of abstraction [27].

Guo, Liang, et al. [80] published a study that minimises the likelihood of data leakage in mobile wireless networks. It illustrates that there is no need to transport data to a cloud server because it trains deep learning models locally, requiring just the transmission of knowledge (model weight or model gradient) rather than data, lowering the risk of data privacy exposure while marginally impairing performance. There have been multiple implementations of modified deep neural networks as well, which can be used to achieve better performance. As a result, deep neural networks in mobile network applications efficiently handle the basic objective of information security.

#### 2.3.5. Deep Belief Network

Deep belief networks are a type of neural network that can be used for unsupervised learning. They are typically composed of multiple layers, and each layer is made up of nodes [27]. The hidden layers in between these two layers contain weights that represent the probability distribution over all possible states at each point in time. These weights can be determined by training or by an unsupervised algorithm called “greedy layer-wise pre-training.” This type of neural network is often trained in a greedy fashion, which means it will start by randomly assigning weights to the nodes and then use unsupervised learning to correct any errors in the weights. The input nodes are connected to each layer in the network. The output layer is then connected to the next layer in the network. The process of unsupervised pre-training is when a deep belief network is trained with input data but without the labels or desired output [53]. 

In mobile networks, Deep Belief Networks can be particularly useful for information management and transmission. Greeshma Arya et al. [81] showed that data transfer in 5G WSN communication may be performed efficiently. The shortest path for packet forwarding has resulted in increased network lifetime and energy efficiency.

#### 2.3.6. Deep Convolutional Neural Network

The deep convolutional neural network consists of one input layer, three convolution and subsampling pairs, three fully linked layers, and a single sigmoid unit in the output layer. The model has proven outstanding accuracy, precision, and F-score for a lot of tasks, along with natural language processing, object recognition, and image classification. The deep convolutional neural network (DCNN) is usually recommended for in-depth malware data processing in mobile networks [82]. Mobile networks can consist of a variety of information in the form of text, images, and videos, all of which can be protected using a DCNN model trained according to the possible vulnerabilities of the network. 

Deep CNN’s incredible learning capacity is because of the utilisation of many features extraction stages that can learn representations from information automatically. Deep CNNs are basically feedforwarding neural networks with BP algorithms implemented to modify the network’s parameters (weights and biases) to minimise the cost of the function’s value. One or more convolutional layers (typically followed by means of a subsampling layer) precede one or more fully related layers within the network [83]. However, many deep CNNs have troubles with overcrowding and require loads of processing time.

#### 2.3.7. Deep Generative Models

Deep generative models based on variational inference allow system behaviours to be learned and attacks to be identified as they occur on their own. This model can handle natural data in its raw form and automatically learn and find its representations, enhancing system knowledge discovery while eliminating the requirement for considerable human engineering and domain expertise [84]. In these models, alternative ways of approximating the input data distribution are learned and then sampled from it to generate previously unknown but credible results [85]. Although deep generative models cannot estimate the density function, they are preferable when it comes to modelling and controlling high-dimensional probability distributions as their training and sampling allow for a better understanding of the underlying complexity. It can be used for inverse reinforcement learning and can be included in reinforcement learning, particularly with generative adversarial networks. The idea is to transform a known and simple distribution (for example, a univariate Gaussian) using a deep neural network that acts as a generator. The features of this model make it feasible for use in the network security domain. 

5G networks that are currently in trend have also implemented deep generative models as in the study presented by Daegyeom Kim et al. [86]. It includes a 5G traffic modelling approach as well as a strategy for determining the required spectrum for privatised 5G networks. By learning from real traffic traces obtained from a large mobile network operator, the generative model is able to produce realistic traffic.

#### 2.3.8. Deep Boltzmann Machine

A deep Boltzmann machine (DBM) is a type of artificial neural network that can be put to use in approximate probability distributions. Boltzmann machines (BM) and restricted Boltzmann machines (RBM) are computational models of neural networks. The BM is a generative model, which means that it generates data samples from a statistical distribution. The RBM is a discriminative model, which means that it can be used to classify data samples. The Boltzmann distribution is a probability distribution that describes the relative likelihood of particles being in one state or another [24]. The Boltzmann machine contains two layers the visible layer and the hidden layer. The visible and hidden layers can have any number of nodes, but each node has an associated weight vector. All nodes are fully connected to every other node in both layers. The connections between nodes in different layers are restricted to within the same layer, so there are no connections between visible and hidden nodes.

DBM has a variety of applications in various domains. Based on a joint deep Boltzmann machine (jDBM) model, Mohammad Rafiqul Alam et al. [87] offer an audio–visual person recognition technique. Tests were conducted on the complex MOBIO database, which contains audio-visual data captured by mobile phones. The results were quite good, demonstrating that the model is resistant to noise and missing data.

#### 2.3.9. Deep Reinforcement Learning

Both reinforcement learning and deep learning are combined in deep reinforcement learning. Reinforcement learning addresses the problem of a computing system being trained to make judgments through trial-and-error. Furthermore, deep reinforcement learning is a deep learning-based system that allows users to make decisions based on unstructured input data without manually engineering the subspace. The model blends high-dimensional problem-solving techniques with reinforcement learning to enable high-dimensional interaction. The fact that an actor is rewarded or penalised based on their actions is intrinsic to this type of machine learning. Actions that lead to the desired result are rewarded (reinforced). A model is trained through trial and error, making this technology suited for dynamic surroundings that vary considerably. 

DRL techniques can be used to cope with growing troubles in communications and networking. The problems embody dynamic community obtaining the right of access to, information rate manipulation, wi-fi caching, facts offloading, community protection, and connectivity protection, which may be all vital to next-generation networks, including 5G and beyond [73]. 

#### 2.3.10. Extreme Learning Machine

The extreme learning machine (ELM) is a single-hidden layer feedforward neural network training algorithm that produces results much quicker than traditional approaches and generates excellent performance [88]. Guang-Bin and Qin-Yu proposed the extreme learning machine (ELM) with the purpose of training single-hidden layer feedforward networks [89]. As ELM learns without the need for an iterative process, it converges much faster than usual algorithms. 

Theoretical investigation revealed that with random parameters, ELM is more likely to obtain a global optimal solution than traditional networks with all the parameters to be trained [90]. ELM is quite popular nowadays because of its variety of applications such as robotics [91], IoT-based models [92,93,94], control systems [93], etc., as well as its high accuracy, cross-domain adaptation, and low time consumption (training time mostly).

Speaking of information security in mobile networks, ELM can be used for localization and positioning systems, data management, analysing mobile signal quality, and many other applications. The model may be used as an efficient and more accurate predictive method for predicting the location of mobile users based on location fingerprint data due to its exceptionally fast processing speed [89]. 

#### 2.3.11. Deep Learning Models for Electronic Information Security

Although complete automation of detection and analysis is a desirable goal, deep learning’s performance in cybersecurity should be evaluated with sensitivity [53]. Repudiation actions, denial of service, information tampering, and leakage are all concerns that electronic information security models for mobile networks face today. Many studies have shown that deep learning technology can assist us in developing the finest security models. DBNs or RBMs or deep autoencoders coupled with classification layers, restricted Boltzmann machines (RBMs), recursive neural networks (RNNs), and several other hybrid DL models have been successfully deployed and have shown promising results in terms of electronic information security, as described in detail by Daniel S. Berman et al. [29]. Improvements in existing DL algorithms will drive research to progress the state of DL in the domain of information security systems for mobile networks. Figure 5 portrays the nomenclature of current deep learning models for electronic information security. Figure 6 depicts the archetypal workflow of machine learning and deep learning models for electronic information security in mobile networks. Figure 7 presents the taxonomy of AI-enabled electronic information security models for mobile networks.

**Table 3 sensors-22-02017-t003:** A summary of works on deep learning models for electronic information security.

Reference	Security-Category	Deep Learning Models Used	Key Contribution	Limitations
[95]	Malware Detection	Deep Convolutional Neural Network (DCNN)	Hand-engineered malware features have no requirement. To make the process easier, the network is trained end-to-end to understand suitable properties and conduct classifications. After the model has been trained, it may be effectively and executed on a GPU with efficiency, permitting a large number of files to be scanned rapidly.	For dynamic and static malware detection on several platforms, it is impractical. Malware detection is incompatible with the design and creation of data augmentation methods.
[96]	Intrusion Detection System (IDS)	Artificial Neural Network (ANN), Stacked Auto Encoder (SAE)	Select the most important features only to reduce their dimensionality. It is suitable for resource-constrained devices. The reduced input features are sufficient for classification tasks.	Limited to lightweight IDS. The issue of a wireless network is difficult to solve.
[97]	Network Traffic Identification	Stacked autoencoder and one-dimensional convolution neural network (CNN)	Both of the tasks such as traffic characterization and application identification are dealt with. Automatic feature extraction saves time and money by eliminating the need for experts to detect and extract handmade elements from traffic, resulting in higher accuracy for traffic classification.	Low efficiency for multi-channel (e.g., differentiating between various types of Skype traffic such as that of chats, video and voice calls) classification and accuracy in classifying Tor’s traffic, etc.
[98]	Spam Email Detection	Bidirectional Encoder Representations from Transformers (BERT)	Effectiveness of word embedding because of hyper-parameter fine-tuning. 98.67% and 98.66% F1 score indicating persistence and robustness of the model.	Smaller input sequence taken. Not valid for text in other languages such as Arabic, etc.
[78]	Intrusion Detection (5G)	RBM; RNN	It can manage traffic fluctuation. Optimising the computational resources at any point in time along with refining the performance and behaviour of analysis and detection procedures is the primary goal. The architecture may adapt and adjust by itself the anomaly detection system depending on the amount of network flows gathered in real-time from 5G subscribers’ user equipment, reducing resource consumption and maximising efficiency.	Because of the abundance of network traffic handled by a RAN, accuracy suffers. Model is not trained for a real-time environment.
[99]	False Data Injection	RBM	The detection scheme is unaffected by the number of attacked data, SVE detection thresholds, and certain degrees of noise in the surroundings. Model can achieve high accuracy for detection in presence of the operation faults occurring now and then.	More realistic FDI attack behaviours are necessary in the model, along with an analysis of the smallest number of sensing units.
[100]	Keystroke Verification	RNN	A high scalability in terms of user count as well as good precision avoiding false positive errors	Takes more time to be fully trained. The classification algorithm selection was affected under the assumption by authors that keystroke dynamics data was sequence-based.
[101]	Border Gateway Protocol Anomaly Detection	RNN	Solve the problem of bursts and noise in dynamic Internet traffic that occur regularly. It learns and grasps traffic patterns using historical features in a sliding time span. The classifier performs well.	It’s vulnerable to overfitting, and using the dropout algorithm to prevent it is challenging. This method is affected by various random weight initialization.
[102]	DGA	CNN RNN	Amenable for real-time detection.	There were 8 DGA that the model was not able to detect.
[103]	Insider Threat	DFNN RNN CNN GNN	DFNN: To detect anomalies one can employ the concept of utilising a deep autoencoder. RNN: Capturing temporal information of the users’ activity sequences. CNN: Great accuracy and precision if the data of a users’ activity can be represented in the form of images. GNN: Organisation information networks are fairly powerful to model the graph data.	Data that is extremely unbalanced. In attacks, there is a lot of temporal information. Fusion of heterogeneous data. There aren’t any practical evaluation metrics. Interpretability. Subtle and Adaptive Threats. Fine-grained Detection.

## 3. Electronic Information Security, Cyber-Attacks and Their Defences

### 3.1. Cyber-Attacks

Any offensive activity that targets electronic information systems, their networks, and infrastructure is referred to as a “cyber-attack”. Its main purpose is to steal, modify, or destroy information. Attack vectors that leverage a lack of readiness and (system as well as human) preparedness to access confidential data or compromise systems are common in the current cyber-attack scenario. Human shortcomings are also used to engineer attack vectors. A cyber-attack can be seen as a situation in which a system or network flaw is exploited by a variety of vulnerabilities. The process of familiarising yourself with new technologies, security trends, and threat intelligence can be a daunting task. Although the target may or may not be aware of all types of cyber-attacks, it might have a mechanism implemented to deal with a few of them. The cause of a cyber-attack may be an inherent risk or a residual risk depending on its risk analysis. Figure 8 depicts the nomenclature of cyber-attacks used in this review. Figure 9 presents the general taxonomy of cyber-threats.

### 3.2. Cyber-Attack Defences

Cyber-attack defences form the basis of cybersecurity and risk management systems. Any information security model is mainly characterised by the way it handles cyber-attack defences. The ever-growing cyberspace is a platform for anyone who shares a network ecosystem through various electronic devices. However, this cyberspace also includes attackers and unauthorised users who have bad intentions. For this reason, it is necessary to have a precautionary approach when dealing with cyber-attack defences. A proper defence mechanism identifies the attack or risk, alerts the system, and functions accordingly to mitigate it. Cyber-attack defence is a concept that can be explained as a set of defined procedures and activities that can be precautionary or come into play during or after a cyber-attack occurs. It checks for signs of a cyber-attack that is either pending, current, or successful. A retrospective analysis can assist us in determining the best cyber-attack defensive method.

### 3.3. Cyber-Attack Datasets

The variety of cyber-attacks has exploded in recent years. This rise in structure and complexity needs more advanced defensive and detecting measures. Traditional ways of identifying cyber-attacks are inefficient, especially in the context of the rising demand for security risks. With the evolution of new technologies such as AI and ML, the datasets that can be used for building complex security systems have also evolved. Some of the old datasets that were quite efficient back then have been improvised now. Moreover, many new datasets have been framed to cope with the existing and upcoming vulnerabilities. Table 4 presents a list of various cyber-attack datasets. 

## 4. Open Problems-Electronic Information Security in Mobile Networks

Recent years have seen a significant increase in the complexity and challenges of managing electronic data. The complexity stems from IS’s ubiquitous and multifunctional character, which attempts to safeguard and rely on information assurance to protect an organisation’s valuable assets while also advancing commercial interactions by creating trust, business alliances, and collaboration platforms [113]. We have identified the following three crucial management issues that need to be addressed in order to fully resolve this issue: (1) compromising system security by addressing risks after the whole system is established; (2) security and information systems are designed in parallel; (3) inadequate thinking is used to shape solutions [114]. Figure 10 illustrates the open problems associated with electronic information security for mobile networks. 

After analysing various electronic information security systems, it can be concluded that the link between the actor and the system needs to be minimised. Here, an actor is referred to as an internal or external entity that interacts with the system. One may also misuse electronic information for political and socio-economic reasons. Such cases can be commonly seen in the medical and political fields because of the personal benefits and greed of an individual. Although transparency in the working environment and user interaction with the system may increase efficiency, it is equally important to ensure a reliable, free from tampering, and fraud-resistant security model. A good information security architecture must look for user mistakes, technical errors, and drawbacks that can act as soft spots for attackers and hackers.

In the real world, we must deal with the challenges of using machine learning models to address a problem in the most efficient manner possible. Models that operate on a single device frequently require data that is too massive for the device to manage. The algorithms might not be able to generalise or scale well in accordance with the complexity of the infrastructure that they run upon. Reducing complexity by designing generalised algorithms that are scalable and don’t have to work with all the data on a single device can help us make more informed decisions with less data, which is good as we will not have to use slower and more resource-intensive methods of training.

The majority of incidents that happened in the last 2–3 years were related to unauthorised network scanning, probing, vulnerable services, viruses, malicious code, and website defacements. Every year, the number of incidents increases, and so does the number of new vulnerabilities. However, they certainly share a certain pattern by which they sophisticatedly attack the system, leaving the user clueless about it. Many of these attacks do not even require new technology to be built to deal with them. Instead, changes in the existing technology can also help in that situation. However, the time available to deal with a threat is so limited that we are sometimes powerless to intervene. 

Any electronic gadget with which we interact nowadays has an interface, which is essentially a software package that contains multiple other software packages. It is imperative to update them regularly. These update distributions may endanger your system. If your company isn’t patching and, as a result, isn’t meeting compliance criteria, regulatory bodies may levy monetary penalties. Patch management should be performed as part of a well-organised, cost-effective, and security-focused procedure [115].

The learning process for deep neural networks often runs into difficulty when the data is unstructured and high-dimensional. Deep learning models require a pre-defined sequence of representations, requiring the data to be fed into the model in an order that can be predicted by the neural network. The pre-defined sequence of representations also limits the complexity of the tasks to be learned. These two limitations make it difficult for deep learning to perform well on structured data and other high-dimensional tasks. Zero-day attack complexities can be another set of risks that exist in the environment when it comes to electronic information systems. 

One of the most well-known and difficult problems with machine learning models is their vulnerability to adversary attacks. These attacks are designed to fool classifiers into misclassifying an input sample as belonging to one class when it actually belongs to another. Current adversarial AI research focuses on techniques in which tiny modifications to ML inputs can fool an ML classifier, causing it to respond incorrectly [116]. Such results prove that these complex attacks are becoming stronger than their defensive countermeasures. These attacks have been observed in many different fields and have been studied extensively, but the security violations they present and their specificity still need more investigation. Recently, some researchers have been focusing on how adversarial scenarios can be handled by AI models. They created a way to build models that are capable of detecting adversarial activities without either leaking information on what they are building or making predictions in an unsafe manner. A scenario-based evaluation of electronic information security models can be the best solution in such situations, but it still poses challenges for the future.

Wireless technology is one of the most successful and booming technologies in the present-day market. Since radio waves can enter through dividers, there is an extraordinary possibility of unapproved admittance to the network and information. Due to the broadcasting nature of wireless signals, anyone can sniff the organisation for its significant accreditations. On the off chance that the organisation isn’t as expected, assailants will obtain adequate information to send off an assault. As a result, the threat is all around us, posing a significant challenge for cyberspace security.

## 5. Future Directions—Electronic Information Security in Mobile Networks

In today’s world, businesses rely on internal computer systems and the Internet to conduct business, and they can’t afford to have their operations disrupted [117]. Not only computer systems but other devices such as smartphones, etc., are significantly affecting network security. Several data analysis activities are performed on remote clouds that are hosted by third parties due to the limited computational capabilities of discrete wireless nodes. Much of the research omits the key step of eliminating privacy-sensitive features from acquired data in lieu of uploading the data to remote clouds.

Privacy has become increasingly important due to the rapid development of technology and the internet. In order for this to not be an issue anymore, there needs to be a greater awareness of the importance of privacy and security. Privacy-preserving encryption schemes can be incorporated as they provide confidentiality protection for messages exchanged between pairs of users using cryptographic keys. Furthermore, privacy- and anonymity-preserving databases and biometric technologies can be embedded together with such models to provide a secure system data storage. This clearly points out the threat to privacy in the domain of big data analytics. Still, a balance needs to be struck between privacy and the quality of data analysis and computational models such as ML algorithms. In terms of security, network topology is equally important. Hence, scalability issues in an interdependent security model can be eliminated in the future to reduce inefficiencies in the model. Economic and privacy concerns lead to under-reporting of security incidents in the security ecosystem [118]. A better technique to cope with networking threats that arise unexpectedly is an area that requires significant improvement. Furthermore, if current models are improvised in certain scenarios, at least crucial information can be kept abstract. In terms of cyber security strategies, AI and ML are the fields that will be utilized in the near future. In such cases, a scenario-based evaluation of electronic information security models may be the best answer, although it still faces future challenges.

Hardware-assisted countermeasures are a new way to protect networks from cyber threats, attacks, and vulnerabilities as they provide an additional layer of protection that is not available with traditional software solutions. These hardware-assisted countermeasures can be classified into the following three categories: memory-centric, data-centric, and device-centric hardware-assisted countermeasures. The most common type of hardware-assisted countermeasure is those that deal with edge computing, or cloud computing, which refers to having data processing performed at the source of the data rather than transferring it over a network connection for processing elsewhere. Wireless networks and cloud storage are also the most accessible and susceptible, making them exposed to a variety of threats such as traffic eavesdropping and capturing, evil twin attacks, brute force attacks, misconfigurations, and so on. Hence, the need for a conceptual approach to trust management can be incorporated into online applications.

There is much ongoing research in quantum computing for machine learning. It is a dynamic field because it has the potential to solve many complex problems and optimise the process of data analysis, but it is also a complex topic because it has many layers and connections to other disciplines such as physics, mathematics, computer science, and so on. The main challenge in quantum computing is the fact that it requires a lot of resources and time for calculations, so it’s not yet ready for commercial use. In the future, quantum computing is expected to be able to solve problems that cannot be solved on classical machines. These problems include optimization problems and solving dynamic systems with many variables. Figure 11 depicts the future research directions associated with electronic information security for mobile networks.

## 6. Conclusions

Early detection and elimination of cyber threats have become of the utmost importance in the contemporary world, be it individuals or mega-organisations dealing with electronic information and data. Adversaries no longer just rely on conventional attack strategies and are evolving over time, which brings about a need for us to develop and evolve the pre-existing defence action plans [119,120,121,122,123,124,125,126,127]. Through this paper, we tend to bring together the various approaches put forward in recent studies through AI-enabled techniques such as ML and DL to enhance a sense of security in mobile networks [128,129,130,131,132,133,134,135,136,137,138,139,140,141,142,143,144,145,146,147,148,149,150,151,152,153,154,155,156,157,158,159,160,161,162,163,164,165,166,167,168,169,170,171,172,173,174,175,176,177,178]. A brief run-through of the popular machine learning algorithms is provided based on the reviewed articles, followed by a survey of proposals tackling numerous security threat categories. The advancements of both cyber-attacks and defence strategies have been organised in this paper; relevant articles have been evaluated to find the impact of different twists introduced in generic ML/DL algorithms on the vulnerabilities faced by electronic information systems and mobile networks. By providing a brief overview of the most recognised datasets that are utilised to train and test the models, a quality breakdown of the cyber-attack datasets has been carried out. Finally, to encourage future researchers and enthusiasts, synopses of current open challenges and potential study areas have been laid out.

## Figures and Tables

**Figure 1 sensors-22-02017-f001:**
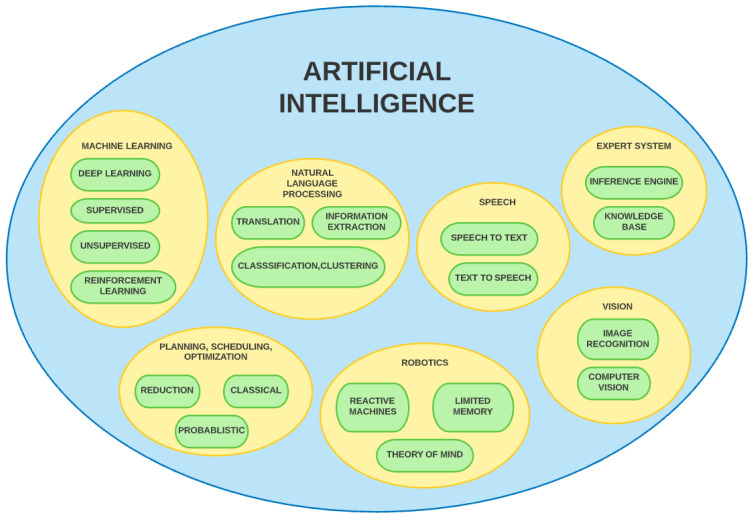
General taxonomy–artificial intelligence techniques.

**Figure 2 sensors-22-02017-f002:**
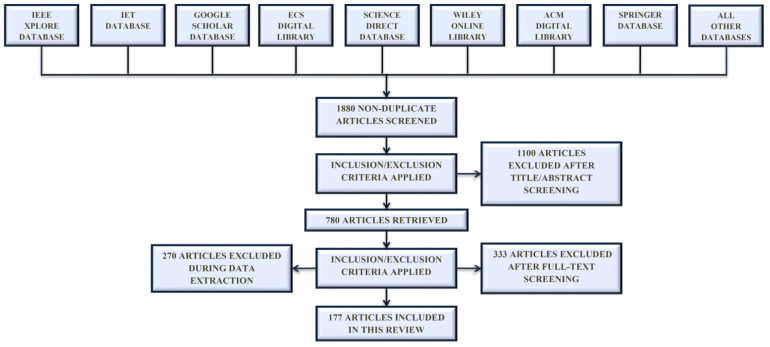
Articles selection process: Machine learning and deep learning models for electronic information security in mobile networks–PRISMA flow chart.

**Figure 3 sensors-22-02017-f003:**
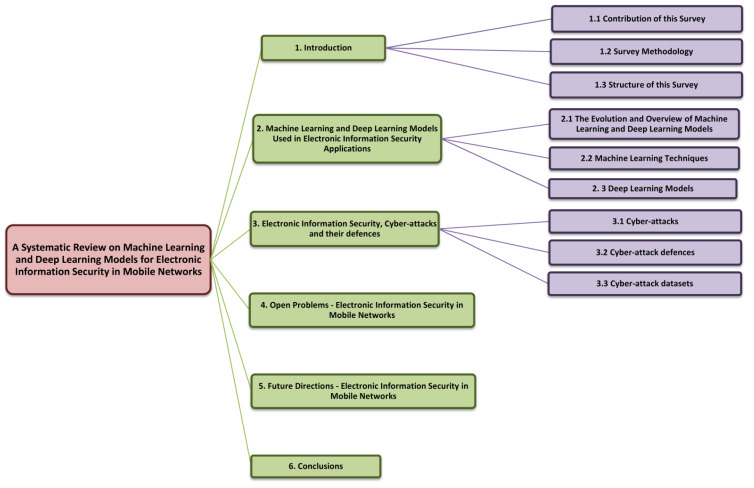
Structural flow of this review.

**Figure 4 sensors-22-02017-f004:**
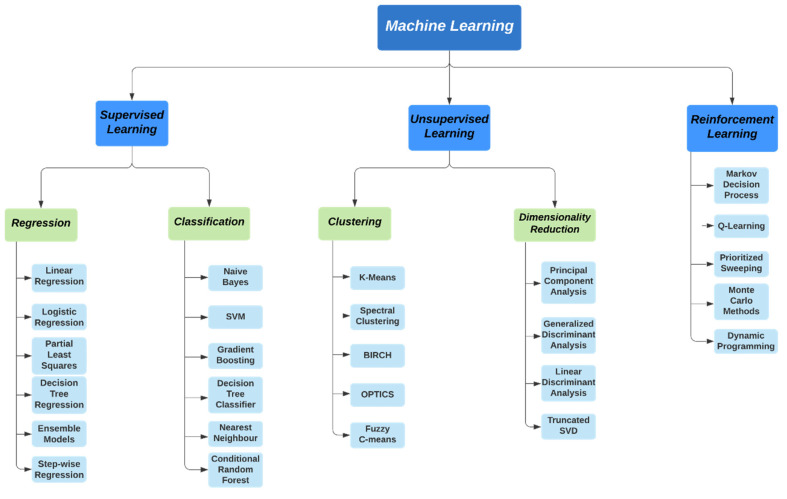
Nomenclature of current machine learning models for electronic information security.

**Figure 5 sensors-22-02017-f005:**
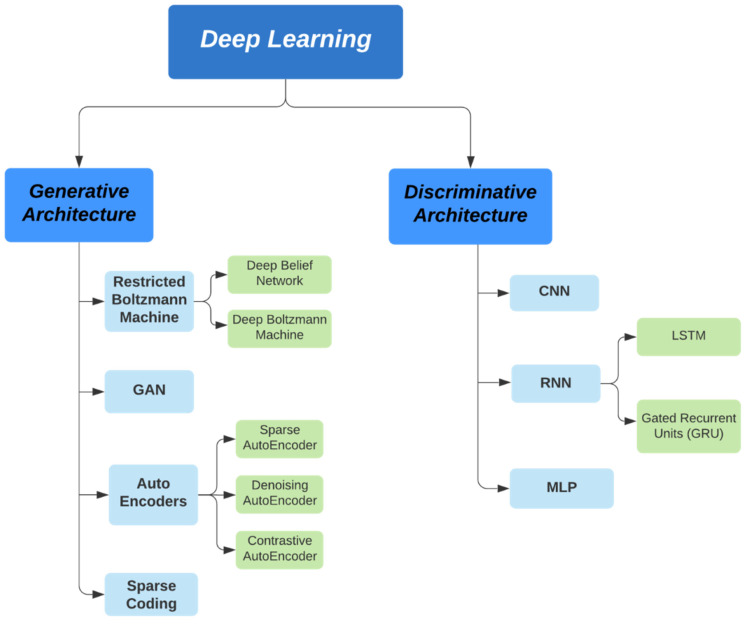
Nomenclature of current deep learning models for electronic information security.

**Figure 6 sensors-22-02017-f006:**
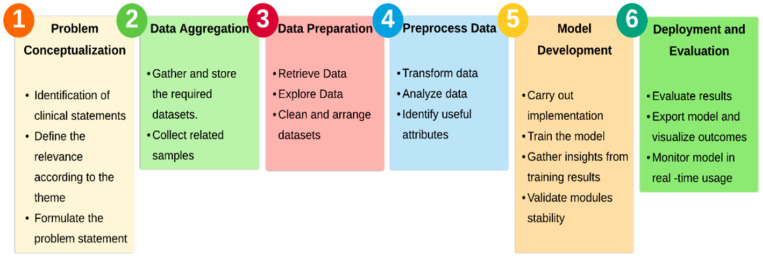
Archetypal workflow of machine learning and deep learning models for electronic information security in mobile networks.

**Figure 7 sensors-22-02017-f007:**
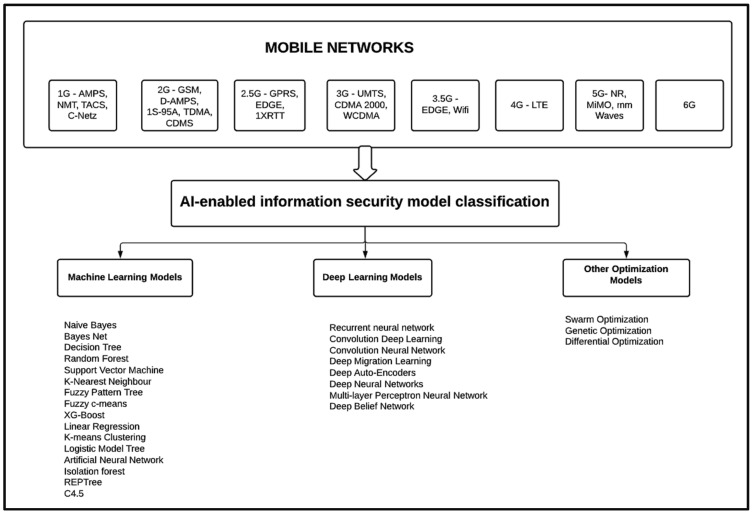
Taxonomy of AI-enabled electronic information security models for mobile networks.

**Figure 8 sensors-22-02017-f008:**
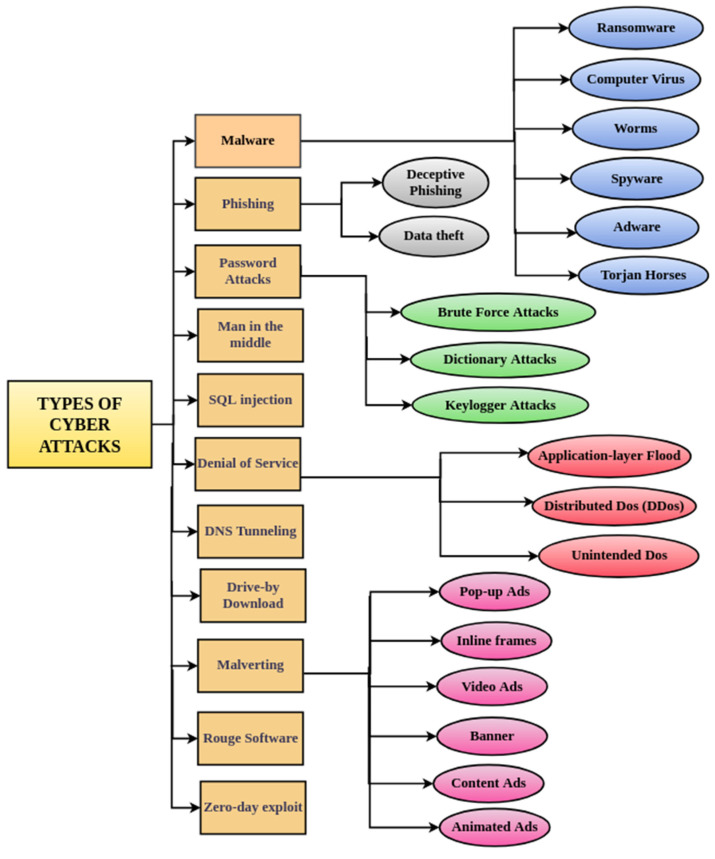
Nomenclature of cyber-attacks used in this review.

**Figure 9 sensors-22-02017-f009:**
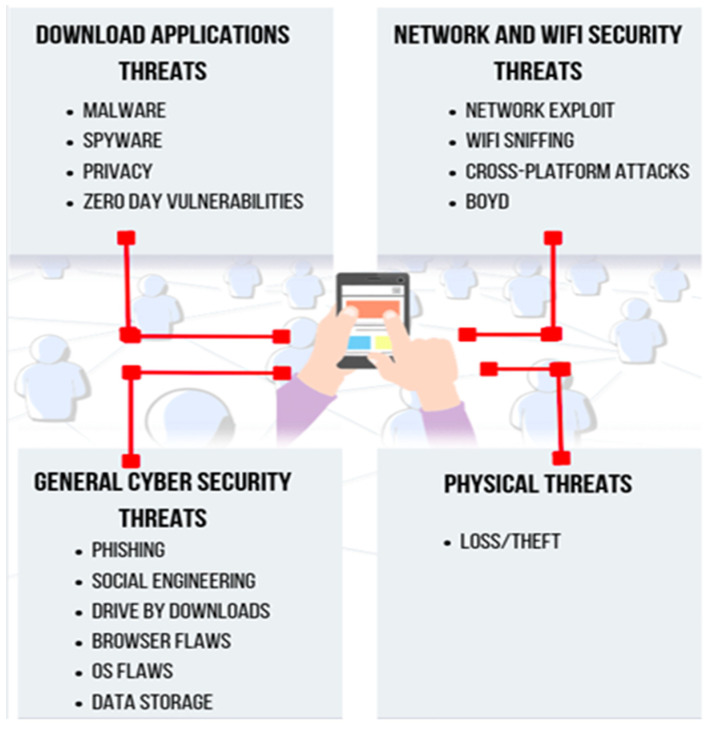
General taxonomy of cyber-threats.

**Figure 10 sensors-22-02017-f010:**
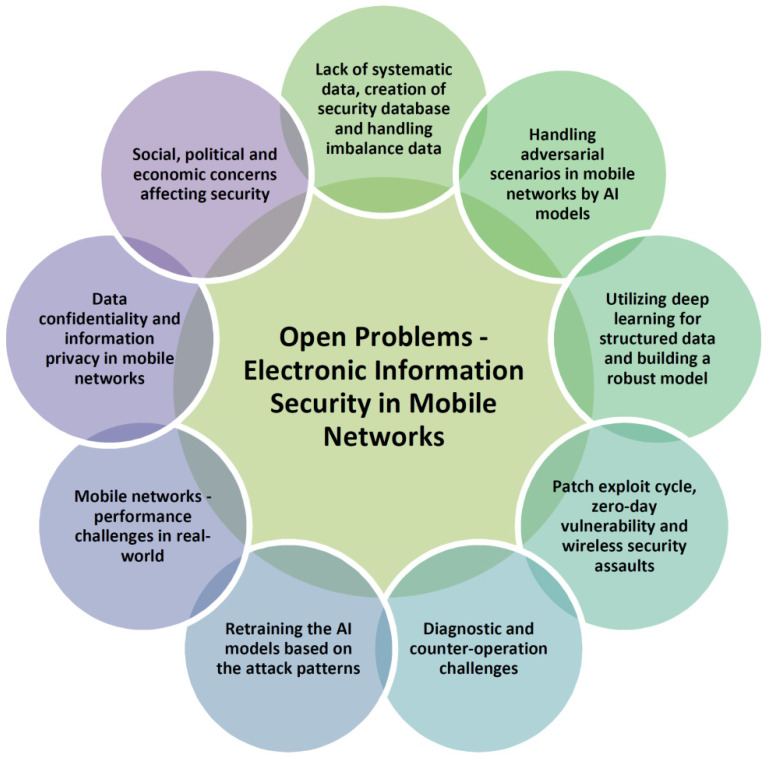
Electronic information security in mobile networks–open problems.

**Figure 11 sensors-22-02017-f011:**
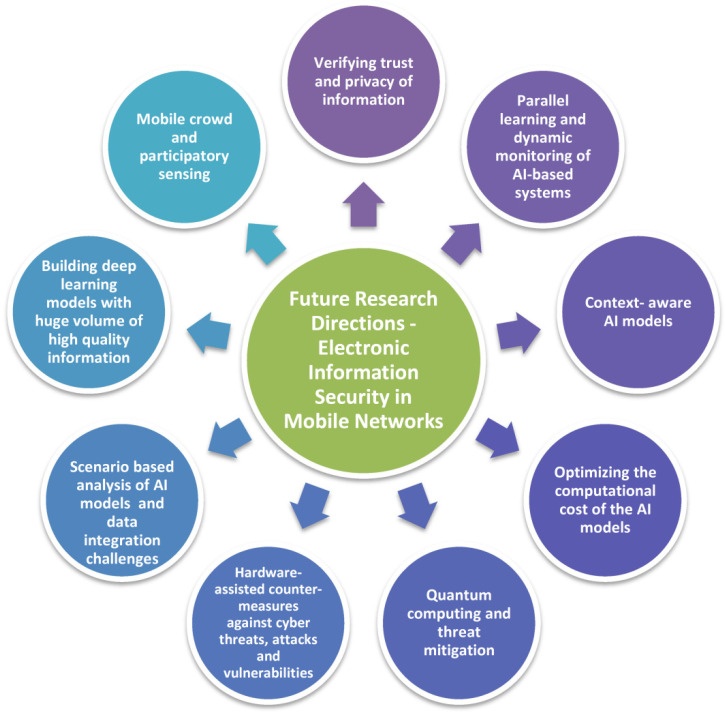
Future research directions—electronic information security in mobile networks.

**Table 1 sensors-22-02017-t001:** Comparison with Previous Reviews. (✓: Discussed, ×: Not Discussed).

Reference	Year	Number of Articles	Time Span	One-Sentence Summary	ML	DL
[16]	2017	260	1986–2017	Intelligent network traffic control systems analysis and future study directions.	✓	✓
[23]	2019	145	2011–2019	UAV communications for 5G networks and upcoming future networks.	×	×
[24]	2019	574	1986–2019	Deep learning techniques in mobile and wireless networks.	✓	✓
[29]	2019	174	1958–2019	Survey on DL methods for cyber security.	×	✓
[33]	2020	65	2004–2020	An examination of AI-enabled phishing attack detection techniques.	✓	✓
[36]	2020	139	1990–2019	Description of several ML approaches used in vehicular networks for communication, networking, and security.	✓	×
[39]	2020	262	2009–2020	ML techniques used for cyber security.	✓	×
[40]	2020	668	1988–2020	ML techniques description and comparison for cyber security.	✓	×
[41]	2020	88	1958–2020	Report on neural networks usage for intrusion detection systems.	✓	✓
[42]	2020	142	1993–2020	Network intrusion detection system.	✓	✓
[43]	2020	175	2002–2020	Survey on moving networks.	×	×
[44]	2020	181	1991–2019	Cyber security data science using machine learning.	✓	×
[50]	2021	189	1989–2021	DL for challenged networks.	×	✓
[51]	2021	138	1999–2020	ML approaches for mobile network and malicious behaviour detection.	✓	×
Our Review	2022	177	1998–2022	This review offers a widespread investigation on the various machine learning and deep learning models for electronic information security in mobile networks.	✓	✓

**Table 2 sensors-22-02017-t002:** A summary of works on machine learning techniques for electronic information security.

Reference	Security-Category	Machine Learning Approaches Used	Key Contribution	Limitations
[63]	Network Attack Patterns	C4.5 Decision Tree; Bayesian Network; Naive-Bayes; Decision Table	Leveraging ML approach for defining security rules on the SDN controller. Viability of ML approach in SDN. Effects of minor security threats on SDN security.	The approach generates variable results for different datasets. A higher variance in data would lead to higher chances of false prediction.
[64]	Network Anomaly Detection	GA; SVM	Select more suitable packet fields through GA using the primary feature selection method. Using the enhanced SVM technique alongside one-class SVM novelty detection ability, enables a high soft margin SVM performance.	A more realistic profiling method would be required to apply the framework in a real TCP/IP traffic environment.
[65]	Traffic Classification	Laplacian SVM	Real-time and adaptive classification of a traffic flow into a QoS category without needing to identify the precise application that originates the traffic flow.	Labelling to be performed explicitly for the datasets in semi-supervised algorithms as unsupervised ML-based algorithms cannot be directly applied in SDN.
[66]	Real-time Intrusion Detection	PSO; SVM	To construct an IDS, an algorithm akin to the PSO-based selection approach is introduced.	Requires improvement in feature selection algorithm on search strategy and evaluation criterion.
[67]	Jamming Attacks	ANN; SVM; LR; KNN; DT; NB	Detection, localization, and avoiding power jamming attacks in optical networks using various ML based solutions. Lowering the probability of successful jamming of lightpaths using resource reallocation scheme that utilises the statistical information of attack detection accuracy.	The studied localization is limited to the jammed channel.
[68]	Malware Detection	DT; NB; RF	Providing a central solution for enterprise security which works on the firewall level in the network. Modern and enhanced machine learning and data mining are used to create a malware detection module.	The proposed solution is not viable for home users, being very processor heavy for a general-purpose machine.
[69]	Network Anomalies (DoS Flooding)	AdaboostM1; RF; MLP	ML related methods are used to detect and classify network intrusions utilising a MIB-based approach. To classify and detect the DoS and brute force attacks use various classifiers. Using ML algorithms for SNMP-MIB data is a very successful strategy for detecting DoS and brute force attacks.	None of the classifiers managed to detect the brute force attack in the TCP dataset.F-Measure results performance is less effective for AdaboostM1 classifiers in the TCP-SYN and UDP flood attacks compared to other attacks.
[70]	Webshell Detection	K-means; MLP; NB; DT; SVM; KNN	For IoT server security experimentation, a new dataset was compiled that included 1551 malicious PHP webshells and 2593 regular PHP scripts. For data pre-processing, term frequency inverse document frequency (TFIDF), opcode, and combined Opcode-TFIDF feature extraction approaches were explored. The dataset is analysed using feature clustering technique based on principal component analysis (PCA). Features important for webshell detection are evaluated.	Tests carried out on machine learning models for webshell detection on PHP scripts only. Higher accuracy results require IoT servers with reliable computing power.
[71]	Jamming-Based Denial-of-Service and Eavesdropping Attacks	MLP; SVM; KNN; DT; Thresh	Proposal of a unique approach to protect wireless communication in a WiNoC from external and internal attackers using persistent jamming-based denial-of-service (DoS) attacks and eavesdropping (ED). Securing communication over wireless channels with a lightweight and low-latency data scrambling mechanism.	In the presence of an internal DoS attack, the performance is not as adequate and only slightly better than a wired NoC.
[72]	Poisoning Attacks (Unreliable Model Updates)	Stochastic Gradient Descent (SGD) Algorithm	Resolving issues of unreliable model updates by introducing reputation as a reliable measure to choose trustworthy workers for reliable federated learning. An effective reputation computation technique is designed using a multi-weight subjective logic model.	Each local worker model trained needs to send regular updates to the central server at regular periods. Insufficient reliable method to monitor worker metrics.

**Table 4 sensors-22-02017-t004:** List of various cyber-attack datasets.

Reference	Year	Dataset Used	Dataset Size	Format	Details about the Dataset/Brief Description
[104]	2016	CAIDA DDoS 2007 and MIT DARPA dataset	5.3 GB	pcap (tcpdump) format	- An hour of anonymized traffic records from a DDoS attack on 4 August 2007 is included in this dataset. - This form of denial-of-service attack tries to prevent users from accessing the server by using all of the computational resources and bandwidth on the network.
[105]	2015	Botnet [Zeus (Snort), Zeus (NETRESEC), Zeus-2 (NIMS), Conficker (CAIDA) and ISOT-Uvic]	14 GB packets	packet	- It is a network-based dataset. It basically works on diverse networks and intercepts emulated traffic (Normal and attack traffic). - The data set is well labelled but not balanced.
[106]	2009	NSL-KDD	4 GB of compressed (approx.)/150k points	tcpdump data	- The train set does not contain any redundant records nor any duplicate records. - A limited number of datasets are taken into consideration for training and testing.
[107]	2011	ISOT	11 GB packets	packet	- The ISOT dataset was compiled from two different datasets comprising malicious traffic from the Honeynet project’s French chapter, which involved the Storm and Waledac botnets, respectively.
[54]	2016	UNSW-NB-15	100 GB	CSV files	- The Australian Centre for Cyber Security’s (ACCS) Cyber Range Lab used an IXIA PerfectStorm programme to construct a combination of realistic modern regular activities and synthetic contemporary attack behaviours from network data.
[108]	2017	Unified Host and Network	150 GB flows (compressed)	bi. flows, logs	- A subset of network and computer (host) events makes up the Unified Host and Network Dataset, events were collected over a 90-day period from the Los Alamos National Laboratory enterprise network.
[109]	2011	Yahoo Password Frequency Corpus	130.64 kB (compressed)	txt files	- The dataset contains sanitised password frequency lists from Yahoo, which were obtained in May 2011.
[110]	2014	500K HTTP Headers	75 MB	CSV files	- Crawled the top 500K sites (as ranked by Alexa).
[111]	2014	The Drebin Dataset	6 MB (approx.)	txt log, CSV and XML files	- The goal of the dataset is to promote Android malware research and allow for comparisons of different detection methods. - There are 5560 applications in the dataset, representing 179 separate malware families. Between August 2010 and October 2012, the samples were collected.
[112]	2008	Common Crawl	320 TiB	WARC and ARC format	- Since 2008, the Common Crawl corpus has accumulated petabytes of data. - Raw web page data, extracted metadata, and text extractions are all included, composed of over 50 billion web pages.

## Data Availability

Not applicable.

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
