# Peer review of "A Systematic Review on Machine Learning and Deep Learning Models for Electronic Information Security in Mobile Networks"

_sensors, 2022, doi:10.3390/s22052017_

Round 1
Reviewer 1 Report
This paper provides the reader with an overview of the security perspective of electronic information. It mainly covers AI-based techniques, security threats and attacks, cyber security datasets, current open challenges and future research directions. The manuscript still has some points which need revision.
- The last paragraph of Section 1. Introduction (before section 1.1) needs to be re-written. The message is not clear and the reader will get confused about the scope of the paper.
- The information provided in Table 1 can be re-arranged, the table in its present form covers a lot of space in comparison to the information being shared in it.
- 2, the WILEY library has to be mistyped as WINLEY, please make the changes.
- Section 2.2.1 says, ANN is two-layered. Please explain how can it be called two-layered if after the next few lines it is stated it has hidden layers also? An input layer, hidden layers, output layer?
- Section 2.2.9, it would be better if the subheading is changed to "machine learning models for electronic information security" from "Other machine learning Models".
- As the previous headings are just the details of the ML models and have no information on how these are used in electronic information security, it would be necessary to add some details on “HOW” here in this subsection 2.2.9.
- I recommend the authors to minimise the number of tables, as they are a lot and misleading.
- Section 1.3 says, “Section I defines the terms "Electronic information Security" and "Mobile networks" and explains how this field has grown in recent years.” In my opinion, the term "Electronic information Security” has been covered and discussed sufficiently in the introduction section, BUT the "Mobile networks" has not been defined properly and discussed adequately. I would say a proper explanation should be given of how you define a mobile network and wireless communication attacks. Some relevant papers given below can be read and cited in the paper:
Author Response
Thank you so much for the positive feedback. We have improved the manuscript based on the reviewer’s comments and suggestions.
Comment 1: The last paragraph of Section 1. Introduction (before section 1.1) needs to be re-written. The message is not clear and the reader will get confused about the scope of the paper.
Response: Thank you for this valuable suggestion. We have re-written and rephrased to make the introduction of the manuscript clear and more comprehendible regarding the scope of the paper.
Comment 2: The information provided in Table 1 can be re-arranged, the table in its present form covers a lot of space in comparison to the information being shared in it.
Response: Thank you for this valuable suggestion. We have re-arranged the table and reduced the space coverage by eliminating some coulmns while keeping the utility of the table unscathed.
Comment 3: the WILEY library has to be mistyped as WINLEY, please make the changes.
Response: Thanks for the correction. We have retyped and used the correct wordings for the same in the revised manuscript.
Comment 4: Section 2.2.1 says, ANN is two-layered. Please explain how can it be called two-layered if after the next few lines it is stated it has hidden layers also? An input layer, hidden layers, output layer?
Response: Thank you for mentioning this point . ANN consists of two visible layers and also incorporate hidden layers within, hence changes have been made in the manuscript addressing the same and clarifying the statement.
Comment 5: Section 2.2.9, it would be better if the subheading is changed to "machine learning models for electronic information security" from "Other machine learning Models".
Response: Thank you for the valuable suggestion. As per the advice, we have changed the subheading to the suggested phrase.
Comment 6: As the previous headings are just the details of the ML models and have no information on how these are used in electronic information security, it would be necessary to add some details on “HOW” here in this subsection 2.2.9.
Response: Thank you for the valuable suggestion. We have included some discussions in each sub-section(2.2.1-2.2.8) by providing relevant results, studies and experimental analysis carried out by fellow researchers as demonstrated in their in published works relevant to our survey. A brief overview of network security is also added as an opener to connect machine learning with network security and further elaborating the section 2.2.9.
Comment 7: I recommend the authors to minimise the number of tables, as they are a lot and misleading.
Response: Thank you for the valuable suggestion. As per your advice we reduced the number of columns in literature survey tables and minimised their coverage space.
Comment 8: Section 1.3 says, “Section I defines the terms "Electronic information Security" and "Mobile networks" and explains how this field has grown in recent years.” In my opinion, the term "Electronic information Security” has been covered and discussed sufficiently in the introduction section, BUT the "Mobile networks" has not been defined properly and discussed adequately. I would say a proper explanation should be given of how you define a mobile network and wireless communication attacks. Some relevant papers given below can be read and cited in the paper.
Response: Thank you for the valuable suggestion. We have included various discussions, results, and experimental analysis covering the mobile network and wireless communication attacks in section 2.2 and 2.3. Relevant papers have been used to support the survery and cited appropriately.
We once again would like to thank the reviewers for their constructive comments that helped to improve the quality of our work. We hope that our response is acceptable for the queries raised by the reviewers.
Thanking you,
Sincerely,
Authors
Reviewer 2 Report
Authors propose a systematic review based on the PRISMA protocol.
The article is very interesting and the content has been clearly described. Moreover, graphical summaries have been used to facilitate the reading and the paper organization.
Prisma workflow has been correctly used and described, and it can be easily recognized by following the paragraphs names
Some minor comments in the following:
- I would suggest to reduce the size of Table 1 by optimizing the empty spaces
- Moreover, please reduce the size of figure 9, since it is not clear
- I would suggest to separate the introduction (paragraph 1) from the subparagraphs 1.1-1.4 that describe how the survey has been carried out. At the end of the introduction a brief description of the following paragraphs must be added. Sub-paragraphs 1.1-1.4 should belong to a proper paragraph, where the review process is described
- Please proofread the article for typos and English grammar and style errors
Author Response
Comment 1: I would suggest to reduce the size of Table 1 by optimizing the empty spaces
Response: Thank you for this valuable suggestion. We have reduced the size of the concerned table by eliminating a couple of columns.
Comment 2: please reduce the size of figure 9, since it is not clear
Response: Thank you for the suggestion. As per the suggestion, we have enhanced the quality of the images and reduced the size to make it more clear.
Comment 3: would suggest to separate the introduction (paragraph 1) from the subparagraphs 1.1-1.4 that describe how the survey has been carried out. At the end of the introduction a brief description of the following paragraphs must be added. Sub-paragraphs 1.1-1.4 should belong to a proper paragraph, where the review process is described
Response: Thank you for the suggestion. We have redone the introduction of our manuscript and incorporated all the suggestions. A brief description of the methodologies of the review process has been provided in the end thus eliminating the need to separate the subparas in a different section of the manuscript.
Comment 4: Please proofread the article for typos and English grammar and style errors
Response: Thank you for the suggestion. We have proofread and updated the manuscript with the correct grammar and styling to the best of our knowledge.
We once again would like to thank the reviewers for their constructive comments that helped to improve the quality of our work. We hope that our response is acceptable for the queries raised by the reviewers.
Thanking you,
Sincerely,
Authors
Reviewer 3 Report
In this review, the authors provide a comprehensive study on several AI-enabled approaches utilized in electronic information security for mobile networks. The article is well organized and the English language is acceptable. Most of the illustrations and collection of the literary works are useful and original. There are some minor issues that the authors need to address due to which I suggest a revision of the paper. In particular, the authors should clarify the following: 1. Authors are suggested to improve the presentation of the abstract - make it succinct and focused. 2. Many abbreviations are used without defination. For example - IT, IEEE, IET, etc., 3. Information about the Open Problems and Future Directions for Electronic Information Security seems to be incomplete. It would be better to modify this section. This review paper must include more content in open problems and future directions as follows: Open Problems: Lack of systematic data, Challenges in utilizing deep learning for structured data, Challenges in Retraining the AI models, Robustness of Deep learning-based security systems, Challenges in Security database creation, Handling Adversarial Scnearios by AI Models, Confidentiality and Protection of Data, Performance challenges in real-world scenario, Challenges in handling imbalance datasets. Authors could include these topics in fig. 10. Also, add a discussion on these topics in sec. 4. Future Directions: Privacy preservation in Electronic Information Security, Hardware-assisted counter-measures against cyber threats, attacks and vulnerabilities, Providing deep learning modles with huge Volume of high quality data, Role of cyber-physical systems, Quantum Computing, Parallel learning of AI Models, Optimizing the computational cost of the AI models, Context-aware AI Models. Authors could include the above mentioned topics in fig. 11. Also, add a discussion on these topics in sec. 5. 4. Authors should follow a standard representation of figures throughout the article. 5. The authors have included reinforcement learning in fig 4. However, the explanation on the utility of reinforcement learning in electronic information the security for mobile networks is missing. 6. In sec. 2.2 and 2.3, authors are expected to add a more focused discussion on the existing utilities/works of each of these algorithms/models corresponding to the several sub-domains in electronic information security for mobile networks. 7.There are minor grammatical and punctuation errors, which should be corrected. 8. Some references are not complete. For example - Ref. 65, authors, journal name, and page details are missing.Author Response
Comment 1: Authors are suggested to improve the presentation of the abstract - make it succinct and focused.
Response: . Thank you for the suggestion. We have worked upon improving the presentation of abstact by giving a brief of our study and including a few discussions on mobile network security concepts thus making it more focused on the topic.
Comment 2: Many abbreviations are used without defination. For example - IT, IEEE, IET, etc.
Response: Thank you for the suggestion. As per the suggestion provided, we have reviewed the manuscript and included the left out abbreviations in the table with their respective definations.
Comment 3: Information about the Open Problems and Future Directions for Electronic Information Security seems to be incomplete. It would be better to modify this section. This review paper must include more content in open problems and future directions as follows: Open Problems: Lack of systematic data, Challenges in utilizing deep learning for structured data, Challenges in Retraining the AI models, Robustness of Deep learning-based security systems, Challenges in Security database creation, Handling Adversarial Scnearios by AI Models, Confidentiality and Protection of Data, Performance challenges in real-world scenario, Challenges in handling imbalance datasets. Authors could include these topics in fig. 10. Also, add a discussion on these topics in sec. 4. Future Directions: Privacy preservation in Electronic Information Security, Hardware-assisted counter-measures against cyber threats, attacks and vulnerabilities, Providing deep learning modles with huge Volume of high quality data, Role of cyber-physical systems, Quantum Computing, Parallel learning of AI Models, Optimizing the computational cost of the AI models, Context-aware AI Models. Authors could include the above mentioned topics in fig. 11. Also, add a discussion on these topics in sec. 5.
Response: Thank you for this valuable suggestion. We have taken the above suggested topics and elaborated more upon their relevance in the current open challenges and scope of those technologies in the upcoming future by giving a brief of each and future directions. Suggestions such as deep learning for structured data, performance challenges in real-world scenarios, handling Adversarial Scnearios by AI Models, etc have been added in Open Problems section. Furthermore in Future Directions section various suggested technologies such as quantum computing, hardware-assisted counter-measures against cyber threats, privacy preserving techniques have been discussed. The diagrams have been recalibrated to include these points. The manuscript has been updated and now provides a better understanding of the content discussed in the paper.
Comment 4: Authors should follow a standard representation of figures throughout the article.
Response: Thank you for the suggestion. As per the suggestion provided, we have changed the representation of some figures to make them more uniform throughout the article.
Comment 5: The authors have included reinforcement learning in fig 4. However, the explanation on the utility of reinforcement learning in electronic information the security for mobile networks is missing.
Response: Thank you for the suggestion. As per the advise given, Reinforcement learning in ML has been added to the section 2.2.9 giving a brief overview of reinforcement learning and how it utilizes self learning to aid in network security.
Comment 6: In sec. 2.2 and 2.3, authors are expected to add a more focused discussion on the existing utilities/works of each of these algorithms/models corresponding to the several sub-domains in electronic information security for mobile networks.
Response: Thank you for the suggestion. As per the suggestion provided, we have extended the previous sections by bringing forth the existing studies and results from the experimental analysis carried out by fellow researchers as demonstrated in their in published works. We have included an ending discussion in every sub section of section 2.2 and 2.3.
Comment 7: There are minor grammatical and punctuation errors, which should be corrected.
Response: Thank you for the suggestion. We have proofread and updated the manuscript with the correct grammar and punctuation to the best of our knowledge.
Comment 8: Some references are not complete. For example - Ref. 65, authors, journal name, and page details are missing.
Response: Thank you for the suggestion. As per the suggestion provided, we have rewritten such references and updated the manuscript.
We once again would like to thank the reviewers for their constructive comments that helped to improve the quality of our work. We hope that our response is acceptable for the queries raised by the reviewers.
Thanking you,
Sincerely,
Authors
Round 2
Reviewer 1 Report
I still find the below citations are worth to be debated and included in the survey paper.
- Defenses against perception-layer attacks on iot smart furniture for impaired people, MM Nasralla, I García-Magariño, J Lloret, IEEE Access 8, 119795-119805
- Internet of Things Based Intelligent Techniques in Workable Computing: An Overview, J Guo, S Nazir, Scientific Programming 2021
- Khan, Muhammad Altaf, et al. "An Efficient Multilevel Probabilistic Model for Abnormal Traffic Detection in Wireless Sensor Networks." Sensors2 (2022): 410.
Author Response
Thank you so much for the positive feedback. We have improved the manuscript based on the reviewer’s comments and suggestions.
Comment 1: I still find the below citations are worth to be debated and included in the survey paper.
- Defenses against perception-layer attacks on iot smart furniture for impaired people, MM Nasralla, I García-Magariño, J Lloret, IEEE Access 8, 119795-119805
- Internet of Things Based Intelligent Techniques in Workable Computing: An Overview, J Guo, S Nazir, Scientific Programming 2021
- Khan, Muhammad Altaf, et al. "An Efficient Multilevel Probabilistic Model for Abnormal Traffic Detection in Wireless Sensor Networks." Sensors2 (2022): 410.
Response: Thank you for this valuable suggestion. We have included these articles in the revised manuscript. Also, based on your earlier suggestion, we have modified the title of our manuscrtipt.
We once again would like to thank the reviewers for their constructive comments that helped to improve the quality of our work. We hope that our response is acceptable for the queries raised by the reviewers.
Thanking you,
Sincerely,
Authors